

# A new integrated rough multi-criteria decision-making model for enterprise resource planning software selection

Bing Cao[1,2], Yongsheng Jin[2], Alptekin Ulutaş[3], Ayse Topal[4], Željko Stević[5,6], Darjan Karabasevic[7] and Cipriana Sava[8]

[1] School of Economics and Management, Dezhou University, Dezhou, China
[2] School of Economics and Management, Beijing University of Posts and Telecommunications, Beijing, China
[3] Department of International Trade and Business, Inonu University, Malatya, Turkey
[4] Department of Business, Nigde Omer Halisdemir University, Nigde, Turkey
[5] Faculty of Transport and Traffic Engineering, University of East Sarajevo, Doboj, Bosnia and Herzegovina
[6] School of Industrial Management Engineering, Korea University, Seoul, Republic of South Korea
[7] Faculty of Applied Management, Economics and Finance, University Business Academy in Novi Sad, Belgrade, Serbia
[8] Faculty of Computers and Applied Informatics, "TIBISCUS" University of Timişoara, Timişoara, Romania

Corresponding authors
Bing Cao, caobing@dzu.edu.cn
Željko Stević,
zeljkostevic88@yahoo.com

## ABSTRACT

Enterprise resource planning (ERP) is widely used to boost the total market power of businesses. The wrong selection is one of the key reasons why ERP installations fail. Due to the complexity of the business environment and the range of ERP systems, choosing an ERP system is a complex and time-consuming procedure. ERP alternatives may be assessed using several criteria, so the ERP selection process may be considered a multi-criteria decision-making (MCDM) problem. In this study, the rough best worst method (BWM) was used to determine criteria weights, while the newly developed rough integrated simple weighted sum product (WISP) was used to rank ERP alternatives. Results suggest that the SFT-4 coded software is regarded as the best option, followed by SFT-5, SFT-6, SFT-2, SFT-3, and SFT-1. Results of the newly developed rough WISP method are compared to those of existing rough techniques in the sensitivity analysis. The differences between them have been found to be negligible. The outcomes show how effectively developed rough BWM and WISP integrated method performs in terms of ERP selection with usability, accuracy, ease of use, and consistency. This study will help decision-makers in a context where ERP is implemented choose the best ERP software for different sectors.

# INTRODUCTION

With the increase of competition in the market, businesses searched for tools to make operations flow more efficiently. Enterprise resource planning (ERP) is one of these tools benefit businesses in terms of efficient operation. ERP is frequently utilized to increase the overall power of enterprises in the market (*Al-Ghofaili & Al-Mashari, 2014*). ERP, including manufacturing resource planning and material requirement planning, is

designed to integrate company operations and services. It encompasses the conventional business operations, such as finance, accounting, sales, human resources, and purchasing. The main goal of using an ERP system, according to *Wallace & Kremzar (2002)*, is to operate the business in a quickly changing and fiercely competitive environment. Even installing an ERP system might be expensive and time-consuming, its advantages are valuable. A business may anticipate gaining major benefits with proper design and choice of the appropriate ERP system, including sharp rises in responses, efficiency, timely deliveries, and revenues, as well as reduction delivery times, prices, quality issues, and inventory.

One of the main causes of ERP installation failure is choosing the wrong system. ERP system selection is a laborious and time-consuming process because of the complexity of the business environment and the variety of ERP solutions. The significance of choosing an appropriate ERP cannot be overstated given the significant financial commitment, possible risks, and advantages since it is a choice on how to form the organizational business (*Haddara, 2018*). Although a variety of techniques have been used to choose an ERP such as mathematical models (*Sagnak & Kazancoglu, 2019*), the usefulness of these techniques is sometimes hindered by complex mathematical models or a lack of sufficient qualities to use in the selection ERP, particularly when some variables are difficult for managers to comprehend and not easily measurable.

The selection process for ERP may be seen as an multi-criteria decision-making (MCDM) problem since each system can be evaluated using a variety of criteria. Therefore, MCDM technologies are frequently employed in ERP selection. MCDM methods are useful tools for choosing the best ERP software from a variety of available options on the market. This study presents a decision model for choosing an ERP based on rough set theory, the simple weighted sum-product (WISP), and the best worst method (BWM) techniques. WISP technique makes it possible to prioritize alternatives much more easily and forecasts the use of a much easier normalization process. It also uses four utility measures to define the total usefulness of the alternatives (*Stanujkic et al., 2021*). By requiring fewer pairwise comparisons, BWM increases the consistency ratio, and because secondary comparisons do not need to be implemented, they are simple and accurate (*Haseli et al., 2021*).

Several extensions of the BWM have been proposed in different fuzzy contexts, which are interval-valued intuitionistic fuzzy (*Dong & Wan, 2024*; *Chen, Wan & Dong, 2023*), intuitionistic fuzzy (*Mohammadi et al., 2022*; *Wan, Dong & Chen, 2024*), triangular fuzzy numbers (TFNs) (*Amiri et al., 2021*; *Ecer & Pamucar, 2020*), D numbers (*Pamučar et al., 2021*), hesitant fuzzy information (*Yang et al., 2020*; *Ali & Rashid, 2019*), probabilistic hesitant fuzzy information (*Li, Wang & Hu, 2019*; *Wang et al., 2022*), neutrosophic numbers (*Pramanik et al., 2023*; *Vafadarnikjoo et al., 2020*), bipolar neutrosophic linguistic numbers (*Nabeeh, Abdel-Monem & Abdelmouty, 2020*; *Edmerdash & Rushdy, 2021*), plithogenic aggregation operations (*Grida, Mohamed & Zaid, 2020*; *Abdel-Basset et al., 2020*).

This study uses the rough set theory as a powerful technique to address uncertainty. The main advantage of rough set theory is that it can handle the complexity and unpredictability of decision-makers from any field without affecting the topic during decision-making compared to fuzzy sets (*Stević et al., 2017b*). Rough set theory has recently been applied to

handle decision problems with conflicting criteria (*Tiwari, Jain & Tandon, 2016*). BWM is a technique developed recently for determining criteria weights in decision-making. It has a few benefits, such as requiring fewer pairwise comparisons than analytic hierarchy process (AHP), yielding more reliable weight coefficients compared to AHP, consistency of results, and using only integer values for pairwise comparisons (*Rezaei, 2015*). The rough BWM method has scarcely been used in the literature. *Pamučar et al. (2017)* used the rough BWM to calculate the weight coefficients of the selection criterion for wind farm locations. *Stević et al. (2017a)* used to calculate the significance of selection criteria for wagons in a logistics firm. *Badi & Ballem (2018)* used rough BWM to determine the weights of the criteria for choosing the best medical supplier. To determine the criteria weights in the location selection problem for construction development, *Stević et al. (2018)* applied the rough BWM. WISP method includes four utility measures for determining the total value of alternatives, forecasts the adoption of a much simpler normalization process, facilitates a much simpler ranking of them, is straightforward to implement and increases judgment dependability (*Stanujkic et al., 2021*). As it is a new method, few studies have used WISP in the literature (*Stanujkić et al., 2021*; *Ulutaş et al., 2022*; *Zavadskas et al., 2022*), and no study uses rough WISP.

This article's contributions to the field can be summarized as follows:

- The selection process incorporated BWM, WISP, rough numbers, and expert judgements. A systematic study was conducted to determine the relationships between ERP requirements and expert opinions.
- This approach can work efficiently in scenarios with uncertain circumstances and complex & limited data, which is a unique feature not found in previous studies.
- The research clearly shows that the suggested method works when applied to issues of ERP selection.
- This study is the first to use BWM and WISP in conjunction with rough numbers; no other research has done so.

The remainder of the article is structured as following: A brief summary of earlier studies about ERP selection is given in the second part. The third part of choosing an ERP presents a unique decision-making framework, including rough BWM and rough WISP methods. The fourth part demonstrates the proposed decision-making technique with a real case example. Eventually, the last part provides the conclusion.

## LITERATURE REVIEW

There are several studies using MCDM methods for decision making problems in the literature. These methods have also been popular for ERP selection and various studies have emerged. We reviewed these studies under three groups: (1) studies using single MCDM method, (2) studies using combined MCDM methods, and (3) studies involving uncertainty in decision making.

The studies using single methods have been summarized as follows: With nine assessment criteria, *Wei, Chien & Wang (2005)* analyzed three potential ERP alternatives using AHP for an electronics business in Taiwan. The analytic network process (ANP) was used with

12 criteria and three alternatives by Perçin (*Perçin, 2008*) as a practical decision-making technique for the ERP selection. *Keçek & Yıldırım (2010)* assessed ERP alternatives with nine criteria and three alternatives for two distinct automotive firms in Turkey using AHP technique. *Armand & Roger (2017)* developed an AHP-based decision framework with five criteria and four alternatives for ERP selection in the case of Cameroon businesses. *Motaki & Kamach (2017)* have used five criteria to evaluate two ERP alternatives with AHP. *Czekster et al. (2019)* used AHP to evaluate two ERP alternatives with nine criteria. *Cruz-Cunha et al. (2021)* used 28 criteria to select the best ERP system between two alternatives for a Portuguese business.

There are few studies using integrated methods in the literature to choose the best ERP for businesses. As the focus has shifted to dealing with uncertainty in recent years, the integrated studies that do not consider uncertainty have been limited. *Kilic, Zaim & Delen (2015)* utilized ANP and Preference Ranking Method for Enrichment Evaluation (PROMETHEE) to select the best ERP system among five alternatives with 11 criteria for small and medium enterprises (SMEs) in Turkey. *López & Ishizaka (2017)* utilized Group AHP Sorting (GAHPSort) and ANP with four criteria and seven alternatives to choose cloud ERP for a business in Spain.

Fuzzy methods have been commonly used to deal with uncertainty in decision problems (*Joshi & Gegov, 2020*; *Joshi, 2019*). *Ayağ & Özdemir (2007)* conducted a case study at a Turkish company in the electronics sector and utilized fuzzy ANP with seven criteria and three alternatives as the approach for choosing ERP software. Using the Fuzzy AHP technique, *Kaur & Mahanti (2008)* analyzed three distinct ERP software solutions based on 14 criteria. Fuzzy ANP was utilized by *Razmi, Sangari & Ghodsi (2009)* to assess the preparedness of an electricity generation business in Iran for ERP deployment. *Kilic, Zaim & Delen (2014)* assessed four ERP alternatives with 12 criteria for the Turkish Airlines Company (THY) using an integrated fuzzy MCDM model including fuzzy logic, AHP and technique for order, and technique for order preference by Similarity to Ideal Solution (TOPSIS) methods. *Hamidi (2015)* used fuzzy AHP with 15 criteria to choose the best ERP system among three alternatives for an Iranian electronics business. *Efe (2016)* used the fuzzy AHP-TOPSIS technique to evaluate four ERP alternatives with 15 criteria, including a Turkish electrical company that manufactures electronic gadgets. *Hinduja & Pandey (2019)* proposed an integrated decision-making framework including Decision-making Trial and Evaluation Laboratory (DEMATEL), intuitionistic fuzzy ANP, and intuitionistic fuzzy AHP to determine which cloud-based ERP solution is ideal for SMEs in India with 19 criteria and 11 alternatives. *Lee, Chen & Kang (2020)* integrated fuzzy set theory, DEMATEL, ANP, and VlseKriterijumska Optimizacija I Kompromisno Resenje in Serbian (VIKOR) to assess four ERP alternatives with five criteria for a company operating in high technology sector. *Garg et al. (2022)* developed a hybrid MCDM technique, including intuitionistic fuzzy soft complex proportional assessment (COPRAS) and Stepwise Weight Assessment Ratio Analysis (SWARA) methods, to assess four ERP alternatives with seven criteria for Indian businesses that produce automobiles.

The literature review indicates that fuzzy methods have been widely used to deal with uncertainty in ERP selection. Unlike fuzzy methods, which rely on an intuitive approach

to determine partial membership, rough numbers establish set borders based on specific parameters. In the context of rough sets, only the arrangement of the input data is utilized without considering additional external criteria. In this article, we developed a decision-making framework allowing the rough BWM and WISP methods to function under unclear circumstances.

The goal of this novel method, which employs rough numbers, is to track changes in the weights of the alternatives in situations including ambiguous information and sparse data. The weights of the decision-makers are determined independently using rough numbers, and the results are carefully contrasted. As a result, rough computation is used to analyze decision-makers' outcomes. By using rough numbers, the need for additional information to determine the uncertainty of the number ranges is eliminated. Compared to the AHP method, BWM allows the weights of criteria and sub-criteria to be obtained easily. The rough BWM makes it possible to take into account doubts that arise during the expert evaluation of the criteria. Additionally, the rough WISP method uses four utility measures. These four utility measures include all arithmetic operations (addition, subtraction, multiplication and division). Therefore, it can be said that the proposed rough WISP method achieves more robust and rigorous results compared to other rough MCDM methods. The rough BWM-WISP integrated MCDM approach has never been used in the existing studies. Methodology in the next section provides a more thorough explanation of the phases of the proposed framework.

## METHODOLOGY

Figure 1 represents an extensive diagram of flow, which is explained mainly in this section and throughout the article.

First of all, the notations utilized for rough set theory are introduced. Let us deliberate that $C$ is the universe encompassing all of the objects, and $D$ denotes an arbitrary object of $(\forall D \in C)$. $A$ includes every object in $C$ encompassing a set of $a$ classes and $A = \{B_1, \ldots, B_q, \ldots, B_a\}$ $(B_q \in A$ and $1 \leq q \leq a)$. These are ordered as $B_1 < \ldots < B_q < \ldots < B_a$ and $A(D)$ demonstrates the class to which object belongs. The upper and lower approximations $(\overline{Apr}(B_q), \underline{Apr}(B_q))$ of class $B_q$ are presented in Eqs. (1) and (2) (*Zavadskas et al., 2018*):

$$\underline{Apr}(B_q) = \{D \in C / A(D) \leq B_q\} \tag{1}$$

$$\overline{Apr}(B_q) = \{D \in C / A(D) \geq B_q\} \tag{2}$$

$B_q$ might be demonstrated as a rough number $(RN(B_q))$ containing upper and lower limits $(\overline{Lim}(B_q), \underline{Lim}(B_q))$ where:

$$\overline{Lim}(B_q) = \frac{1}{N^u} \sum_{D \in \overline{Apr}(B_q)} A(D) \tag{3}$$

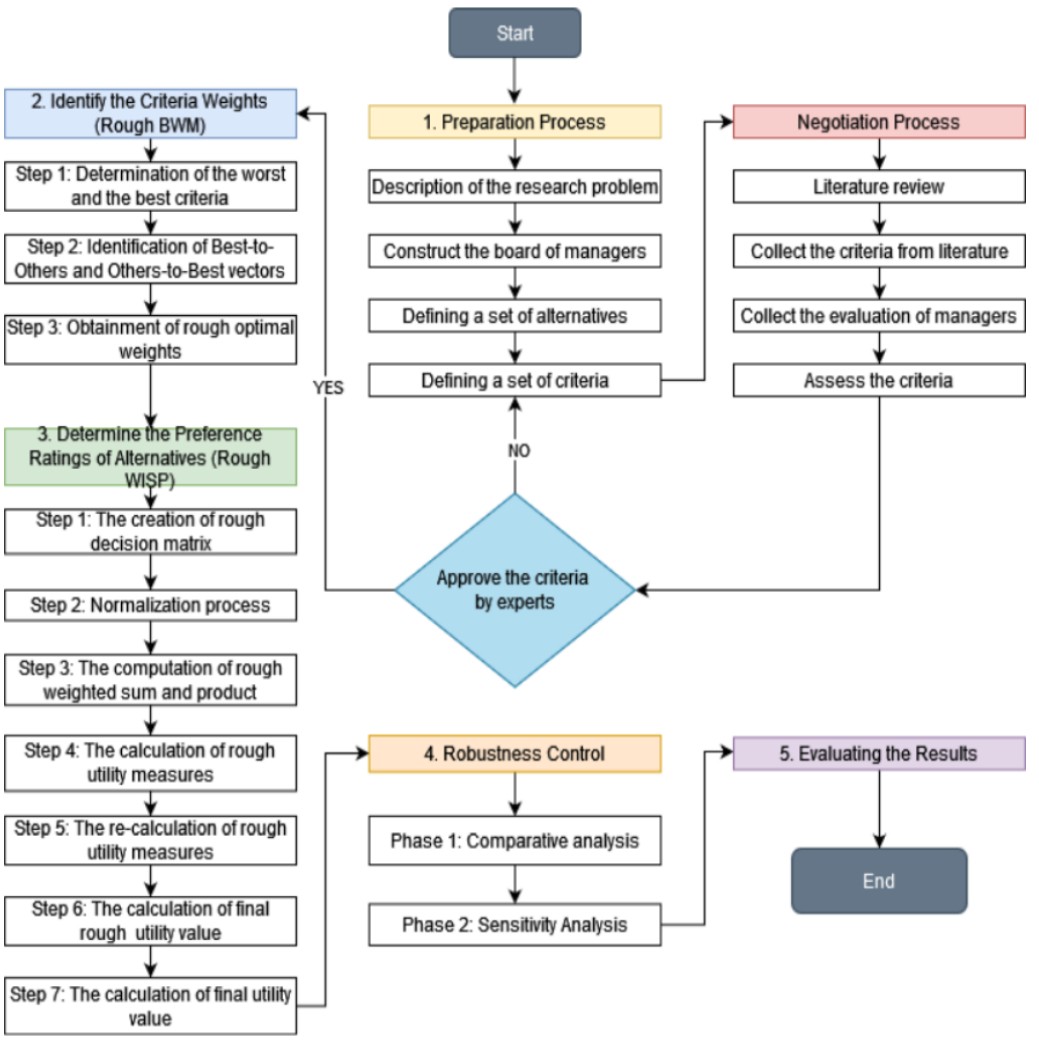

**Figure 1   Diagram of research.**

$$\underline{Lim}(B_q) = \frac{1}{N^l} \sum_{D \in \underline{Apr}(B_q)} A(D) \tag{4}$$

$$RN(B_q) = \left[ \underline{Lim}(B_q), \overline{Lim}(B_q) \right] \tag{5}$$

where $N^u$ and $N^l$ are the numbers of objects comprised in $\overline{Apr}(B_q)$ and $\underline{Apr}(B_q)$, respectively.

The arithmetic operations between the rough numbers ($RN(E)$ and $RN(Z)$) defined above are shown below.

Addition:

$$RN(E) + RN(Z) = \left[ \underline{Lim}(E) + \underline{Lim}(Z), \overline{Lim}(E) + \overline{Lim}(Z) \right] \tag{6}$$

Subtraction:

$$RN(E) - RN(Z) = \left[ \underline{Lim}(E) - \overline{Lim}(Z), \overline{Lim}(E) - \underline{Lim}(Z) \right] \qquad (7)$$

Multiplication:

$$RN(E) \times RN(Z) = \left[ \underline{Lim}(E) \times \underline{Lim}(Z), \overline{Lim}(E) \times +\overline{Lim}(Z) \right] \qquad (8)$$

Division:

$$RN(E) \div RN(Z) = \left[ \underline{Lim}(E) \div \overline{Lim}(Z), \overline{Lim}(E) \div \underline{Lim}(Z) \right] \qquad (9)$$

In the next subsections, the methodologies of the rough BWM and rough WISP methods will be illustrated.

## Rough BWM

In this study, the weights of the ERP selection criteria will be determined using the rough BWM method. The steps of this method are summarized below (*Lo et al., 2019*; *Chang et al., 2019*).

Step 1: ERP selection criteria are identified.

Step 2: The worst and the best criteria are determined.

Step 3: Best-to-Others and Others-to-Worst Vectors are identified.

Each decision-maker assesses the relative importance $F_{BEj}^{(r)}$ of the best criterion $BE$ to other criteria $j$ to determine the best-to-others ($BEO$) vector with using 1–9 scale.

$$F_{BEj}^{(r)} = \left( f_{BE1}^{(r)}, f_{BE2}^{(r)}, \ldots, f_{BEBE}^{(r)}, \ldots, f_{BEn}^{(r)} \right) \qquad (10)$$

Likewise, each decision-maker assesses the relative importance $F_{jW}^{(r)}$ of other criteria $j$ to the worst criterion $W$ to obtain the others-to-worst ($OW$) vector with using 1–9 scale.

$$F_{jW}^{(r)} = \left( f_{1W}^{(r)}, f_{2W}^{(r)}, \ldots, f_{WW}^{(r)}, \ldots, f_{nW}^{(r)} \right) \qquad (11)$$

In Eqs. (10) and (11), $f_{BEBE}^{(r)}$ and $f_{WW}^{(r)}$ equal to 1.

Rough $BEO$ and $OW$ vectors are calculated by using Eqs. (3) and (4). In other words, the $BEO$ and $OW$ vectors of each decision maker are combined with Eqs. (3) and (4). These rough vectors are shown in Eqs. (12) and (14).

$$RN(F_{BEj}) = \left[ f_{BEj}^l, f_{BEj}^u \right] = \left( [f_{BE1}^l, f_{BE1}^u], [f_{BE2}^l, f_{BE2}^u], \ldots, [f_{BEBE}^l, f_{BEBE}^u], \ldots, [f_{BEn}^l, f_{BEn}^u] \right) \qquad (12)$$

$$RN(F_{jW}) = \left[ f_{jW}^l, f_{jW}^u \right] = \left( [f_{1W}^l, f_{1W}^u], [f_{2W}^l, f_{2W}^u], \ldots, [f_{WW}^l, f_{WW}^u], \ldots, [f_{nW}^l, f_{nW}^u] \right) \qquad (13)$$

Step 4: The rough optimal weights $(RN\left(w_j^*\right)=([w_1^{l*},w_1^{u*}],[w_2^{l*},w_2^{u*}],\ldots,[w_n^{l*},w_n^{u*}]))$ are computed by Eq. (14).

$$Min\varepsilon^*s.t.\begin{cases} w_{BE}^l - f_{BEj}^l.w_j^u \le \varepsilon^*.w_j^u \\ w_{BE}^l - f_{BEj}^l.w_j^u \ge -\varepsilon^*.w_j^u \\ w_{BE}^u - f_{BEj}^u.w_j^l \le \varepsilon^*.w_j^l \\ w_{BE}^u - f_{BEj}^u.w_j^l \ge -\varepsilon^*.w_j^l \\ w_j^l - f_{jW}^l.w_W^u \le \varepsilon^*.w_W^u \\ w_j^l - f_{jW}^l.w_W^u \ge -\varepsilon^*.w_W^u \\ w_j^u - f_{jW}^u.w_W^l \le \varepsilon^*.w_W^l \\ w_j^u - f_{jW}^u.w_W^l \ge -\varepsilon^*.w_W^l \\ \sum_{j=1}^{n}\left(\dfrac{w_j^l+w_j^u}{2}\right)=1 \\ w_j^u \ge w_j^l \ge 0 \end{cases} \tag{14}$$

In Eq. (14), $\varepsilon^*$ indicates the consistency ratio ($CR$) of the pairwise comparison matrix. If this value is near 0, that means the pairwise comparison matrix is consistent.

**Rough WISP**

In this study, the rough WISP method is used to rank ERP software. The steps of rough WISP are shown below.

Step 1: Each decision-makers assess alternatives based on selection criteria with using 1–9 scale. A rough decision matrix ($RN(G)$) is created by using Eqs. (3) and (4). $RN(G)$ is shown in Eq. (15).

$$RN(G)=\left[g_{ij}^l,g_{ij}^u\right]_{m\times n} \tag{15}$$

Step 2: This rough decision matrix is normalized by Eq. (16) to structure rough normalized matrix ($RN(H)=\left[h_{ij}^l,h_{ij}^u\right]$).

$$RN(H)=\left[h_{ij}^l,h_{ij}^u\right]=\left[\frac{h_{ij}^l}{max(h_{ij}^u)},\frac{h_{ij}^u}{max(h_{ij}^u)}\right] \tag{16}$$

Step 3: The rough weighted sum and weighted product of normalized values are computed for each alternative with respect to non-beneficial ($NBL$) and beneficial ($BFL$) criteria.

$$RN\left(K^+\right)=\left[k_{ij}^{l+},k_{ij}^{u+}\right]=\sum_{j\in BFL}\left[h_{ij}^l\times w_j^l,h_{ij}^u\times w_j^u\right] \tag{17}$$

$$RN\left(K^-\right)=\left[k_{ij}^{l-},k_{ij}^{u-}\right]=\sum_{j\in NBL}\left[h_{ij}^l\times w_j^l,h_{ij}^u\times w_j^u\right] \tag{18}$$

$$RNRN\left(P^+\right)=\left[p_{ij}^{l+},p_{ij}^{u+}\right]=\prod_{j\in BFL}\left[h_{ij}^l\times w_j^l,h_{ij}^u\times w_j^u\right] \tag{19}$$

$$RN\left(P^-\right) = \left[p_{ij}^{l-}, p_{ij}^{u-}\right] = \prod_{j \in NBL}\left[h_{ij}^l \times w_j^l, h_{ij}^u \times w_j^u\right] \tag{20}$$

Step 4: The rough utility measures are obtained as follows.

$$RN\left(Y_i^{sd}\right) = \left[y_i^{sdl}, y_i^{sdu}\right] = \left[k_{ij}^{l+} - k_{ij}^{u-}, k_{ij}^{u+} - k_{ij}^{l-}\right] \tag{21}$$

$$RN\left(Y_i^{td}\right) = \left[y_i^{tdl}, y_i^{tdu}\right] = \left[p_{ij}^{l+} - p_{ij}^{u-}, p_{ij}^{u+} - p_{ij}^{l-}\right] \tag{22}$$

$$RN\left(Y_i^{sr}\right) = \left[y_i^{srl}, y_i^{sru}\right] = \left[\frac{k_{ij}^{l+}}{k_{ij}^{u-}}, \frac{k_{ij}^{u+}}{k_{ij}^{l-}}\right] \tag{23}$$

$$RN\left(Y_i^{tr}\right) = \left[y_i^{trl}, y_i^{tru}\right] = \left[\frac{p_{ij}^{l+}}{p_{ij}^{u-}}, \frac{p_{ij}^{u+}}{p_{ij}^{l-}}\right] \tag{24}$$

Step 5: The rough utility measures are re-computed as follows.

$$RN\left(\underline{Y}_i^{sd}\right) = \left[\underline{y}_{-i}^{sdl}, \underline{y}_{-i}^{sdu}\right] = \left[\frac{1 + y_i^{sdl}}{1 + max\left(y_i^{sdu}\right)}, \frac{1 + y_i^{sdu}}{1 + max\left(y_i^{sdu}\right)}\right] \tag{25}$$

$$RN\left(\underline{Y}_i^{td}\right) = \left[\underline{y}_{-i}^{tdl}, \underline{y}_{-i}^{tdu}\right] = \left[\frac{1 + y_i^{tdl}}{1 + max\left(y_i^{tdu}\right)}, \frac{1 + y_i^{tdu}}{1 + max\left(y_i^{tdu}\right)}\right] \tag{26}$$

$$RN\left(\underline{Y}_i^{sr}\right) = \left[\underline{y}_{-i}^{srl}, \underline{y}_{-i}^{sru}\right] = \left[\frac{1 + y_i^{srl}}{1 + max\left(y_i^{sru}\right)}, \frac{1 + y_i^{sru}}{1 + max\left(y_i^{sru}\right)}\right] \tag{27}$$

$$RN\left(\underline{Y}_i^{tr}\right) = \left[\underline{y}_{-i}^{trl}, \underline{y}_{-i}^{tru}\right] = \left[\frac{1 + y_i^{trl}}{1 + max\left(y_i^{tru}\right)}, \frac{1 + y_i^{tru}}{1 + max\left(y_i^{tru}\right)}\right] \tag{28}$$

Step 6: The final rough utility value is calculated for each alternative by Eq. (29).

$$RN\left(Y_i\right) = \left[y_i^l, y_i^u\right] = \frac{1}{4} \times \left(\left[\left(\underline{y}_i^{sdl} + \underline{y}_i^{tdl} + \underline{y}_i^{srl} + \underline{y}_i^{trl}\right), \left(\underline{y}_i^{sdu} + \underline{y}_i^{tdu} + \underline{y}_i^{sru} + \underline{y}_i^{tru}\right)\right]\right) \tag{29}$$

Step 7: The final rough utility values are transformed into crisp utility values by Eq. (30).

$$Y_i = \frac{y_i^l + y_i^u}{2} \tag{30}$$

**Table 1  Main criteria and sub-criteria.**

| Main criteria | Sub-criteria |
| --- | --- |
| Costs (C) | Improvement Costs (IC) |
| | Purchasing Costs (PC) |
| | Maintenance and Support Costs (MSC) |
| User-Related Features (URF) | Ease of Use (EU) |
| | Reporting Skills (RS) |
| | Software Firm's Reputation (SFR) |
| | Quality of Support Services (QSS) |
| | Functionality (F) |
| System Structure Related Features (STRF) | Ease of Integration into the System (EIS) |
| | System Reliability (SR) |
| | Cross Module Integration (CMI) |
| | Compliance with System (CS) |

**Table 2  The best-to-others (*BEO*) vector and rough numbers.**

| Managers | Best | STRF | URF | C |
| --- | --- | --- | --- | --- |
| Mang-1 | C | 3 | 2 | 1 |
| Mang-2 | C | 4 | 2 | 1 |
| Mang-3 | C | 4 | 3 | 1 |
| Mang-4 | C | 3 | 2 | 1 |
| Mang-5 | C | 4 | 3 | 1 |
| Mang-6 | C | 4 | 2 | 1 |
| Mang-7 | C | 4 | 3 | 1 |
| Rough numbers | C | [3.510, 3.918] | [2.184, 2.674] | [1.000, 1.000] |

## APPLICATION

This study was carried out in a textile company. The company would like to better organize its corporate structure by purchasing ERP software. The company has worked with ERP software before; however, it was not satisfied. Therefore, this ERP software company is not included in the list of alternatives. The ERP software to be purchased was determined by interviewing the seven senior managers of the company. Managers identified six ERP software as alternatives. The managers determined three main criteria and twelve sub-criteria for the selection criteria. The main criteria and sub-criteria are shown in Table 1.

Of the criteria shown in Table 1, only three are *NBL* criteria, and the rest are *BFL* criteria. The *NBL* criteria are as follows: IC, PC and MSC. Managers first determined the worst and best criteria. Then, each decision maker determined the importance of the best criterion relative to the other criteria and the importance of the other criteria relative to the worst criterion with a 1–9 scale. In order to make the information obtained from managers more detailed and comprehensive, the 1–9 scale was used in this study. With the aid of Eq. (9) and (10), the vectors BEO and OW are determined. Table 2 shows the *BEO* vector and the rough numbers for the main criteria. Table 3 demonstrates the *OW* vector and the rough numbers for the main criteria.

**Table 3  The others-to-worst (*OW*) vector and rough numbers.**

| Managers | Best | STRF | URF | C |
|---|---|---|---|---|
| Mang-1 | STRF | 1 | 2 | 3 |
| Mang-2 | STRF | 1 | 3 | 4 |
| Mang-3 | STRF | 1 | 3 | 4 |
| Mang-4 | STRF | 1 | 3 | 3 |
| Mang-5 | STRF | 1 | 3 | 4 |
| Mang-6 | STRF | 1 | 2 | 4 |
| Mang-7 | STRF | 1 | 4 | 4 |
| Rough numbers | STRF | [1.000, 1.000] | [2.504, 3.216] | [3.510, 3.918] |

**Table 4  The rough weights of the main criteria.**

| Main criteria | Rough weights | $\varepsilon^\star$ |
|---|---|---|
| STRF | [0.128, 0.137] | |
| URF | [0.268, 0.340] | 0.549 |
| C | [0.557, 0.570] | |

**Table 5  The local and global rough weights of the sub-criteria.**

| Main criteria | Main criteria' rough weights | Sub-Criteria | Sub-criteria' local rough weights | Sub-criteria' global rough weights | $\varepsilon^\star$ |
|---|---|---|---|---|---|
| STRF | [0.128, 0.137] | EIS | [0.200, 0.222] | [0.026, 0.030] | 0.675 |
| | | SR | [0.407, 0.443] | [0.052, 0.061] | |
| | | CMI | [0.096, 0.097] | [0.012, 0.013] | |
| | | CS | [0.244, 0.289] | [0.031, 0.040] | |
| URF | [0.268, 0.340] | EU | [0.342, 0.375] | [0.092, 0.128] | 0.781 |
| | | RS | [0.253, 0.276] | [0.068, 0.094] | |
| | | SFR | [0.050, 0.053] | [0.013, 0.018] | |
| | | QSS | [0.119, 0.125] | [0.032, 0.043] | |
| | | F | [0.184, 0.221] | [0.049, 0.075] | |
| C | [0.557, 0.570] | IC | [0.113, 0.113] | [0.063, 0.064] | 0.552 |
| | | PC | [0.544, 0.609] | [0.303, 0.347] | |
| | | MSC | [0.287, 0.333] | [0.160, 0.190] | |

The rough weights of the main criteria are identified by placing these obtained rough numbers into the linear program indicated in Eq. (14). The rough weights of the main criteria are indicated in Table 4.

The local rough weights of the sub-criteria are determined by performing the same processing processes for the sub-criteria. Then, the rough weights of the main criteria and the local rough weights of the sub-criteria are multiplied using Eq. (8) and the global rough weights of the sub-criteria are obtained. The local and global rough weights of the sub-criteria are shown in Table 5.

Using Eq. (30), interval values are converted into crisp values and criteria and sub-criteria are ranked according to their importance values. The main criteria are ranked according

to their importance as follows: C (0.564), URF (0.304) and STRF (0.133). In the same way, this equation is used for the sub-criteria and the importance ranking of the sub-criteria is obtained. According to the results obtained, the sub-criteria are listed as follows. PC (0.325), MSC (0.175), EU (0.110), RS (0.081), IC (0.064), F (0.062), SR (0.057), QSS (0.038), CS (0.036), EIS (0.028), SFR (0.016) and CMI (0.013). The most important main criterion in ERP selection was determined as cost (C). C criterion is followed by user-related features (URF) criterion. The least important criterion is determined as System Structure Related Features (STRF) criterion. As can be seen, criterion C is determined as the most important criterion in ERP selection. The three most important sub-criteria are as follows: PC, MSC and EU. As can be seen, purchasing cost (PC) is the most important sub-criterion, followed by another cost sub-criterion, maintenance and support costs (MSC). The third most important sub-criterion is ease of use (EU), which is a sub-criterion of the URF main criterion. The least important sub-criterion is the CMI sub-criterion, which is a sub-criterion of the STRF main criterion. According to the results, criterion C is much more important than the other main criteria and the sub-criteria of this main criterion, PC and MSC, are also very important among the sub-criteria. From this result, it can be understood that the Cost criterion is very important for this company and that purchasing cost and maintenance and support costs are of vital importance for this company. Only one sub-criterion (EU) of the URF main criterion was included in the top three sub-criteria, while none of the sub-criteria of the STRF main criterion was included in the top three. After the ranking of the criteria and sub-criteria is finished, the computation is continued with the rough weights before using Eq.(30). ERP software was evaluated after the rough weights of the sub-criteria are determined . For this, each manager evaluates the performance of the alternatives in the criteria by using a scale of 1–9. After the evaluations of the managers, these evaluations are combined with Eq. (3) and Eq. (4) to obtain rough numbers and a rough decision matrix consisting of these rough numbers is formed. This rough decision matrix including ERP software (SFT) alternatives and selection criteria is indicated in Table 6.

With Eq. (16), the values in the rough decision matrix are normalized. Table 7 presents these normalized values.

Equations (17)–20) are used to compute the rough weighted sum and weighted product of normalized values for each alternative with respect to non-beneficial and beneficial criteria. The rough utility measures are calculated using Eq. (21)–(24). Equations (25)–(28) are used to re-compute rough utility measures. All these results are presented in Table 8.

Equation (29) is utilized to compute final rough utility value ($RN(Y_i)$) for each alternative. Then, these rough values are converted into crisp utility values by Eq. (30). Table 9 presents these results and the rankings of software.

According to Table 9, while the best software is determined as SFT-4 coded software, this software is followed by SFT-5, SFT-6, SFT-2, SFT-3, and SFT-1 coded software, respectively. The results of the developed rough WISP method are checked through comparative analysis (*Švadlenka et al., 2023*) with other rough MCDM methods (rough measurement of alternatives and ranking according to compromise solution (MARCOS) (*Stević et al., 2023*), rough TOPSIS (*Xuan et al., 2022*), rough additive ratio assessment (ARAS) (*Radović*

**Table 6   The rough decision matrix.**

| Criteria alternatives | EIS | SR | CMI | CS |
|---|---|---|---|---|
| SFT-1 | [7.367, 8.347] | [8.082, 8.490] | [5.633, 7.255] | [5.979, 7.449] |
| SFT-2 | [5.306, 7.265] | [5.979, 7.449] | [3.500, 6.393] | [6.612, 7.944] |
| SFT-3 | [3.367, 4.347] | [5.367, 6.347] | [5.163, 5.979] | [5.551, 7.021] |
| SFT-4 | [4.306, 6.265] | [5.660, 7.510] | [7.082, 7.490] | [6.326, 6.816] |
| SFT-5 | [5.367, 6.347] | [5.367, 6.347] | [7.020, 7.265] | [6.184, 6.674] |
| SFT-6 | [4.327, 5.959] | [4.384, 5.891] | [5.082, 5.694] | [4.327, 5.959] |
| **Criteria alternatives** | **EU** | **RS** | **SFR** | **QSS** |
| SFT-1 | [3.959, 6.898] | [8.326, 8.816] | [7.784, 8.496] | [8.082, 8.490] |
| SFT-2 | [7.326, 7.816] | [5.604, 7.337] | [5.204, 7.577] | [6.306, 8.265] |
| SFT-3 | [4.718, 6.735] | [4.859, 6.014] | [3.706, 5.535] | [5.316, 6.127] |
| SFT-4 | [5.354, 6.987] | [4.784, 5.496] | [3.978, 6.839] | [4.735, 6.694] |
| SFT-5 | [6.184, 6.674] | [4.859, 6.014] | [4.633, 7.082] | [5.306, 7.265] |
| SFT-6 | [3.633, 5.255] | [4.784, 5.496] | [4.245, 5.469] | [5.163, 5.979] |
| **Criteria alternatives** | **F** | **IC** | **PC** | **MSC** |
| SFT-1 | [8.082, 8.490] | [4.870, 7.469] | [8.326, 8.816] | [8.184, 8.674] |
| SFT-2 | [5.000, 7.086] | [2.782, 4.335] | [6.020, 6.265] | [6.021, 6.837] |
| SFT-3 | [5.326, 5.816] | [3.337, 4.439] | [5.082, 5.490] | [6.082, 6.490] |
| SFT-4 | [4.564, 6.535] | [3.833, 5.277] | [3.326, 3.816] | [5.510, 5.918] |
| SFT-5 | [4.306, 6.265] | [4.469, 6.053] | [3.704, 4.850] | [5.326, 5.816] |
| SFT-6 | [5.082, 5.490] | [5.048, 6.895] | [3.184, 3.674] | [4.367, 5.347] |

*et al., 2018*), rough multi-objective optimization method on the basis of ratio analysis (MOORA) (*Zaher, Khalifa & Mohamed, 2018*), rough COPRAS (*Pamučar et al., 2018*), rough simple additive weighting (SAW) (*Durmić et al., 2020*) and rough combinative distance-based assessment (CODAS) (*Regaieg Cherif & Moalla Frikha, 2021*). Table 10 indicates these comparison results.

According to Table 10, while the rough WISP, rough TOPSIS and rough SAW methods achieve the same results, there is little difference between the results of the other rough MCDM methods and the results of the rough WISP method. All of the eight rough MCDM methods have identified the ERP software coded SFT-4 in the 1st place. In addition, the Pearson correlation coefficient between the rough WISP method and other rough MCDM methods (rough MARCOS, rough ARAS, rough COPRAS, rough MOORA and rough CODAS) has been calculated as 0.943. Because of all these, it is concluded that the developed rough WISP method reaches the accurate results.

Each rough MCDM method achieves good results in its own way. However, the rough WISP method uses four utility measures. These utility measures also include four arithmetic operations. Therefore, it can be said that the proposed rough WISP method achieves more rigorous and robust results.

**Table 7  The rough normalized decision matrix.**

| Criteria Alternatives | EIS | SR | CMI | CS |
|---|---|---|---|---|
| SFT-1 | [0.883, 1.000] | [0.952, 1.000] | [0.752, 0.969] | [0.753, 0.938] |
| SFT-2 | [0.636, 0.870] | [0.704, 0.877] | [0.467, 0.854] | [0.832, 1.000] |
| SFT-3 | [0.403, 0.521] | [0.632, 0.748] | [0.689, 0.798] | [0.699, 0.884] |
| SFT-4 | [0.516, 0.751] | [0.667, 0.885] | [0.946, 1.000] | [0.796, 0.858] |
| SFT-5 | [0.643, 0.760] | [0.632, 0.748] | [0.937, 0.970] | [0.778, 0.840] |
| SFT-6 | [0.518, 0.714] | [0.516, 0.694] | [0.679, 0.760] | [0.545, 0.750] |
| **Criteria alternatives** | **EU** | **RS** | **SFR** | **QSS** |
| SFT-1 | [0.507, 0.883] | [0.944, 1.000] | [0.916, 1.000] | [0.952, 1.000] |
| SFT-2 | [0.937, 1.000] | [0.636, 0.832] | [0.613, 0.892] | [0.743, 0.973] |
| SFT-3 | [0.604, 0.862] | [0.551, 0.682] | [0.436, 0.651] | [0.626, 0.722] |
| SFT-4 | [0.685, 0.894] | [0.543, 0.623] | [0.468, 0.805] | [0.558, 0.788] |
| SFT-5 | [0.791, 0.854] | [0.551, 0.682] | [0.545, 0.834] | [0.625, 0.856] |
| SFT-6 | [0.465, 0.672 | [0.543, 0.623] | [0.500, 0.644] | [0.608, 0.704] |
| **Criteria alternatives** | **F** | **IC** | **PC** | **MSC** |
| SFT-1 | [0.952, 1.000] | [0.652, 1.000] | [0.944, 1.000] | [0.944, 1.000] |
| SFT-2 | [0.589, 0.835] | [0.372, 0.580] | [0.683, 0.711] | [0.694, 0.788] |
| SFT-3 | [0.627, 0.685] | [0.447, 0.594] | [0.576, 0.623] | [0.701, 0.748] |
| SFT-4 | [0.538, 0.770] | [0.513, 0.707] | [0.377, 0.433] | [0.635, 0.682] |
| SFT-5 | [0.507, 0.738] | [0.598, 0.810] | [0.420, 0.550] | [0.614, 0.671] |
| SFT-6 | [0.599, 0.647] | [0.676, 0.923] | [0.361, 0.417] | [0.503, 0.616] |

## Sensitivity analysis

In this analysis, a total of thirty scenarios were arranged by reducing the weights of the three criteria (PC, EU and MSC) with the highest weights using the following method. Equation (31), which is used to arrange the scenarios, is presented below (*Huskanović, Stević & Simić, 2023*; *Badi & Elghoul, 2023*).

$$W_{n\gamma} = (1 - W_{n\theta}) \frac{W_\gamma}{(1 - W_n)} \tag{31}$$

In Eq. (31), $W_{n\gamma}$ is a new value of the weight of criterion, additionally, $W_\gamma$ indicates original value of criterion. Besides, $W_{n\theta}$ presents the reduced criterion weight, and $W_n$ is the original weight of the criterion with a reduced value (*Tešić et al., 2023*; *Wieckowski et al., 2023*). Figure 2 indicates the results of the sensitivity analysis.

Criteria weights were changed with 30 scenarios. SFT-3 coded ERP software kept its 5th place in all scenarios. Other ERP software has changed places at least once. ERP software with SFT-1 code was ranked 4th only in S10 scenario. In the remaining scenarios, it took the 6th place. ERP software with SFT-2 code was ranked 3rd in S3 and S4, 2nd in S5 and S6, and 1st in S7-S10. In other scenarios, it maintained its place in the 4th place. SFT-4 coded ERP software was ranked 2nd in S7-0-S10 and ranked 1st in other scenarios. While the SFT-5 coded ERP software was in the 3rd place in S5–S10 and S17–S20 scenarios, it

**Table 8  The results of the rough BWM-WISP model.**

| Results Alternatives | $RN(K^+)$ | $RN(K^-)$ | $RN(P^+)$ | $RN(P^-)$ |
|---|---|---|---|---|
| SFT-1 | [0.305, 0.484] | [0.478, 0.601] | [1.21301E−14, 5.33435E−13] | [0.001774577, 0.00421952] |
| SFT-2 | [0.275, 0.457] | [0.341, 0.434] | [1.78376E−15, 2.61131E−13] | [0.00053855, 0.00137116] |
| SFT-3 | [0.223, 0.376] | [0.315, 0.396] | [4.48065E−16, 3.45851E−14] | [0.000551253, 0.00116799] |
| SFT-4 | [0.234, 0.403] | [0.248, 0.325] | [8.68637E−16, 1.03111E−13] | [0.00037509, 0.00088096] |
| SFT-5 | [0.247, 0.395] | [0.263, 0.370] | [1.4301E−15, 9.44711E−14] | [0.000471001, 0.00126134] |
| SFT-6 | [0.200, 0.339] | [0.232, 0.321] | [2.91679E−16, 2.3054E−14] | [0.000374908, 0.00100042] |
| **Results alternatives** | $RN(\overline{Y_i^{sd}})$ | $RN(\overline{Y_i^{td}})$ | $RN(\overline{Y_i^{sr}})$ | $RN(\overline{Y_i^{tr}})$ |
| SFT-1 | [−0.296, 0.006] | [−0.00421952, −0.00177457] | [0.507488, 1.012552] | [2.87477E−12, 3.00599E−10] |
| SFT-2 | [−0.159, 0.116] | [−0.00137115, −0.00053855] | [0.633641, 1.340176] | [1.30092E−12, 4.84878E−10] |
| SFT-3 | [−0.173, 0.061] | [−0.00116799, −0.00055125] | [0.563131, 1.193651] | [3.83621E−13, 6.27391E−11] |
| SFT-4 | [−0.091, 0.155] | [−0.00088095, −0.00037509] | [0.72, 1.625] | [9.86015E−13, 2.74896E−10] |
| SFT-5 | [−0.123, 0.132] | [−0.00126134, −0.00047100] | [0.667568, 1.501901] | [1.13379E−12, 2.00575E−10] |
| SFT-6 | [−0.121, 0.107] | [−0.00100041, −0.00037490] | [0.623053, 1.461207] | [2.91557E−13, 6.14925E−11] |
| **Results alternatives** | $RN(\underline{Y_i^{sd}})$ | $RN(\underline{Y_i^{td}})$ | $RN(\underline{Y_i^{sr}})$ | $RN(\underline{Y_i^{tr}})$ |
| SFT-1 | [0.60952, 0.870996] | [0.996154, 0.9986] | [0.574281, 0.766687] | [1.000, 1.000] |
| SFT-2 | [0.72813, 0.966234] | [0.999003, 0.999836] | [0.622339, 0.891496] | [1.000, 1.000] |
| SFT-3 | [0.71601, 0.918615] | [0.999207, 0.999824] | [0.595479, 0.835676] | [1.000, 1.000] |
| SFT-4 | [0.78701, 1.000] | [0.999494, 1.000] | [0.655238, 1.000] | [1.000, 1.000] |
| SFT-5 | [0.75930, 0.980087] | [0.999113, 0.999904] | [0.635264, 0.953105] | [1.000, 1.000] |
| SFT-6 | [0.76103, 0.958442] | [0.999374, 1.000] | [0.618306, 0.937603] | [1.000, 1.000] |

**Table 9  The rankings of the rough BWM-WISP model.**

| Results alternatives | $RN(Y_i)$ | $Y_i$ | Rankings |
|---|---|---|---|
| SFT-1 | [0.79499, 0.909071] | 0.8520 | 6 |
| SFT-2 | [0.83737, 0.964391] | 0.9009 | 4 |
| SFT-3 | [0.827676, 0.938529] | 0.8831 | 5 |
| SFT-4 | [0.860436, 1.0000] | 0.9302 | 1 |
| SFT-5 | [0.848421, 0.983274] | 0.9158 | 2 |
| SFT-6 | [0.84468, 0.974011] | 0.9093 | 3 |

kept its second place in other scenarios. In addition, in other scenarios, it kept its 3rd place. It can be said that the rough WISP method developed according to the results is sensitive to the change in the weights of the criteria.

WS (*Sałabun & Urbaniak, 2020*) and SCC (*Wieckowski et al., 2023*) coefficients can be calculated if there are changes in the results verification analysis. Calculated SCC (Table 11)

**Table 10  The comparison of rough MCDM methods.**

| Methods Alternatives | Rough WISP | Rough MARCOS | Rough ARAS | Rough COPRAS | Rough TOPSIS | Rough MOORA | Rough CODAS | Rough SAW |
|---|---|---|---|---|---|---|---|---|
| SFT-1 | 6 | 6 | 6 | 6 | 6 | 6 | 6 | 6 |
| SFT-2 | 4 | 4 | 4 | 4 | 4 | 4 | 4 | 4 |
| SFT-3 | 5 | 5 | 5 | 5 | 5 | 5 | 5 | 5 |
| SFT-4 | 1 | 1 | 1 | 1 | 1 | 1 | 1 | 1 |
| SFT-5 | 2 | 3 | 3 | 3 | 2 | 3 | 3 | 2 |
| SFT-6 | 3 | 2 | 2 | 2 | 3 | 2 | 2 | 3 |

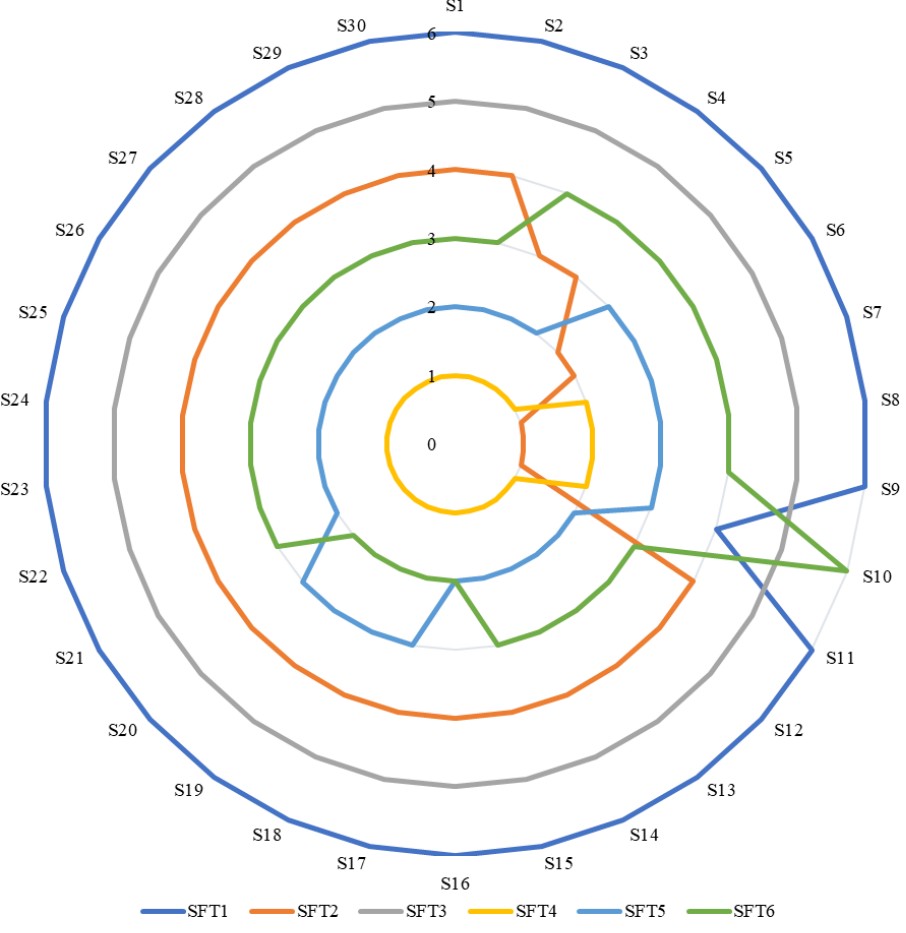

**Figure 2  Results of sensitivity analysis.**

and WS (Table 12) coefficients for comparative analysis and for sensitivity analysis (Fig. 3) are as follows:

All calculated statistical correlation coefficients show high and very high correlations between initial rank and others obtained through comparative and sensitivity analysis.

**Table 11  Spearman's rank correlation coefficient for ranks in comparative analysis.**

| SCC | Rough WISP | Rough MARCOS | Rough ARAS | Rough COPRAS | Rough TOPSIS | Rough MOORA | Rough CODAS | Rough SAW | AV |
|---|---|---|---|---|---|---|---|---|---|
| Rough WISP | 1.000 | 0.943 | 0.943 | 0.943 | 1.000 | 0.943 | 0.943 | 1.000 | 0.964 |
| Rough MARCOS | 0.943 | 1.000 | 1.000 | 1.000 | 0.943 | 1.000 | 1.000 | 0.943 | 0.979 |
| Rough ARAS | 0.943 | 1.000 | 1.000 | 1.000 | 0.943 | 1.000 | 1.000 | 0.943 | 0.979 |
| Rough COPRAS | 0.943 | 1.000 | 1.000 | 1.000 | 0.943 | 1.000 | 1.000 | 0.943 | 0.979 |
| Rough TOPSIS | 1.000 | 0.943 | 0.943 | 0.943 | 1.000 | 0.943 | 0.943 | 1.000 | 0.964 |
| Rough MOORA | 0.943 | 1.000 | 1.000 | 1.000 | 0.943 | 1.000 | 1.000 | 0.943 | 0.979 |
| Rough CODAS | 0.943 | 1.000 | 1.000 | 1.000 | 0.943 | 1.000 | 1.000 | 0.943 | 0.979 |
| Rough SAW | 1.000 | 0.943 | 0.943 | 0.943 | 1.000 | 0.943 | 0.943 | 1.000 | 0.964 |

**Table 12  WS correlation coefficient for ranks in comparative analysis.**

| WS | Rough WISP | Rough MARCOS | Rough ARAS | Rough COPRAS | Rough TOPSIS | Rough MOORA | Rough CODAS | Rough SAW | AV |
|---|---|---|---|---|---|---|---|---|---|
| Rough WISP | 1.000 | 0.896 | 0.896 | 0.896 | 1.000 | 0.896 | 0.896 | 1.000 | 0.935 |
| Rough MARCOS | 0.896 | 1.000 | 1.000 | 1.000 | 0.896 | 1.000 | 1.000 | 0.896 | 0.961 |
| Rough ARAS | 0.896 | 1.000 | 1.000 | 1.000 | 0.896 | 1.000 | 1.000 | 0.896 | 0.961 |
| Rough COPRAS | 0.896 | 1.000 | 1.000 | 1.000 | 0.896 | 1.000 | 1.000 | 0.896 | 0.961 |
| Rough TOPSIS | 1.000 | 0.896 | 0.896 | 0.896 | 1.000 | 0.896 | 0.896 | 1.000 | 0.935 |
| Rough MOORA | 0.896 | 1.000 | 1.000 | 1.000 | 0.896 | 1.000 | 1.000 | 0.896 | 0.961 |
| Rough CODAS | 0.896 | 1.000 | 1.000 | 1.000 | 0.896 | 1.000 | 1.000 | 0.896 | 0.961 |
| Rough SAW | 1.000 | 0.896 | 0.896 | 0.896 | 1.000 | 0.896 | 0.896 | 1.000 | 0.935 |

## DISCUSSION

This study focuses on the selection of the best ERP software to improve the operational efficiency of companies in an environment of intense market competition. One of the most important obstacles to the successful implementation of ERP software is undoubtedly its compatibility with the company's system. Companies want to work with the best ERP software that can integrate with their systems. More than one criterion should be considered when determining ERP software.

Recognizing the challenges posed by the complex nature of ERP software selection, this study advocates the use of MCDM methods. These techniques play an invaluable role in overcoming the complexity of the decision-making process by providing a systematic and rational approach to selecting the most appropriate ERP software. In particular, this study proposes a new decision model that integrates rough set theory, WISP, and BWM approaches to facilitate the selection process.

The decision model presented in this study introduces a new approach to ERP software selection by utilizing rough WISP and rough BWM. The results of the study reveal that SFT-4 emerges as the highest scoring alternative, followed by SFT-5, SFT-6, SFT-2, SFT-3 and SFT-1. This ranking is a proof of the effectiveness of the proposed decision model in distinguishing the most suitable ERP software among the available alternatives.

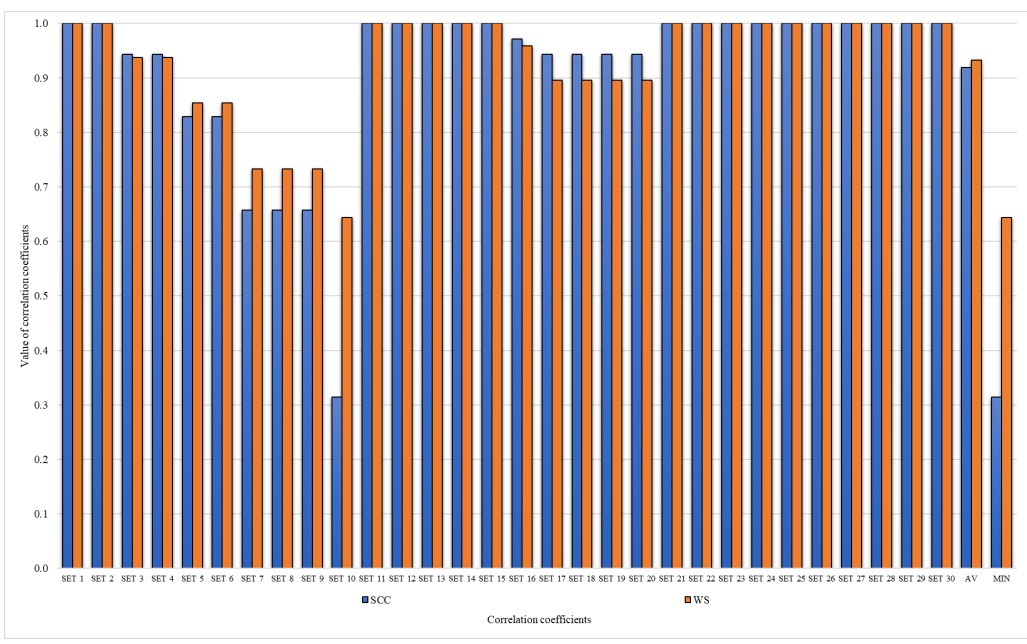

**Figure 3  WS and SCC correlation coefficients for ranks in sensitivity analysis.**

A critical evaluation of the rough WISP method's results in comparison to other established rough MCDM approaches, including rough MARCOS, rough TOPSIS, rough ARAS, rough MOORA, rough COPRAS, rough SAW, and rough CODAS, reveals noteworthy findings. While rough WISP, rough TOPSIS, and rough SAW methods yield same results, little variations are observed in the outcomes of other rough MCDM methods. Importantly, the Pearson correlation coefficient of 0.943 between the rough WISP method and other rough MCDM methods underscores the accuracy and reliability of the proposed approach.

Despite the contributions of this study, certain limitations must be acknowledged. The reliance solely on subjective (expert opinions) data, the absence of objective data, and the omission of objective weighting methods like preference selection index (PSI), statistical variance (SV), or criteria importance through intercriteria correlation (CRITIC) are notable shortcomings. Additionally, the evaluation of only three main criteria and twelve sub-criteria may limit the generalizability of the findings.

To address the limitations identified, future research should aim to incorporate objective data, employ objective weighting methods, consider a broader range of criteria, and explore extensions of the WISP technique. The study suggests potential applications of the rough WISP method in addressing various MCDM problems beyond ERP software selection, extending its utility to different sectors.

In conclusion, this study presents a comprehensive rough MCDM model for ERP software selection and introduces the rough WISP method as a valuable contribution. Despite its limitations, the study opens avenues for future research, promising a more

thorough understanding of ERP system selection processes and the broader applicability of the proposed decision model.

## CONCLUSION

Companies looked for techniques to improve the efficiency of their operations as market competitiveness increased. The incorrect system selection is one of the key reasons why ERP installations go wrong. The complexity of the company environment and the wide range of ERP systems make choosing an ERP system a tedious and time-consuming procedure. With so many alternatives on the market, MCDM techniques are helpful tools for selecting the finest ERP software. This article proposes a novel decision model for selecting an ERP based on rough set theory, WISP, and BWM approaches. According to the results, the SFT-4 coded software is rated as the best, followed by SFT-5, SFT-6, SFT-2, SFT-3, and SFT-1 coded software. The created rough WISP method's results are compared with those of existing rough MCDM approaches (rough MARCOS, rough TOPSIS, rough ARAS, rough MOORA, rough COPRAS, rough SAW and rough CODAS). While the rough WISP, rough TOPSIS, and rough SAW methods achieve the same results, there is little difference between the results of the other rough MCDM methods and the results of the rough WISP method. The ERP program with the code SFT-4 has been found by all MCDM approaches. Additionally, it has been determined that the rough WISP technique and other rough MCDM methods (rough MARCOS, rough ARAS, rough COPRAS, rough MOORA and rough CODAS) approaches have a Pearson correlation coefficient of 0.943. It may be inferred from all of them that the created rough WISP approach yields accurate results.

Although this study provided a comprehensive rough MCDM model, including rough BWM and rough WISP, it also has a number of shortcomings. Firstly, no objective data were used in this research, and only subjective (expert opinions) data were utilized. Future studies will be able to offer more thorough and useful conclusions by using the factories' objective data. Furthermore, no objective weighting methods like PSI, SV, or CRITIC were used in this study. Future research can employ one of these techniques to build a stronger model. Only subjective data were used in this study; historical data were excluded. Future studies will be able to offer thorough results by using previous data. In this study, only three main criteria and twelve sub-criteria were evaluated. Future studies may consider many main and sub-criteria. The stochastic, neutrosophic, and plithogenic extensions of the WISP approach have not yet been developed due to the recent development of the WISP technique. Consequently, these WISP technique enhancements could be explored in future research. Additionally, they can use the rough WISP method developed in this study to handle different MCDM problems (third-party logistics provider selection, supplier selection, machine selection, *etc.*). The developed rough WISP method can also be used in future studies to address the issue of ERP software selection in other sectors (machinery, chemical, automotive, *etc.*).

### Funding
The authors received no funding for this work.

### Competing Interests
Zeljko Stević is an Academic Editor for PeerJ.

### Author Contributions
- Bing Cao conceived and designed the experiments, analyzed the data, performed the computation work, authored or reviewed drafts of the article, and approved the final draft.
- Yongsheng Jin performed the experiments, analyzed the data, performed the computation work, authored or reviewed drafts of the article, and approved the final draft.
- Alptekin Ulutaş conceived and designed the experiments, performed the experiments, analyzed the data, performed the computation work, prepared figures and/or tables, authored or reviewed drafts of the article, and approved the final draft.
- Ayse Topal conceived and designed the experiments, performed the experiments, analyzed the data, performed the computation work, prepared figures and/or tables, and approved the final draft.
- Željko Stević performed the experiments, analyzed the data, prepared figures and/or tables, and approved the final draft.
- Darjan Karabasevic conceived and designed the experiments, performed the experiments, analyzed the data, authored or reviewed drafts of the article, and approved the final draft.
- Cipriana Sava analyzed the data, prepared figures and/or tables, and approved the final draft.

### Data Availability
  The raw data and measurements are available in the Supplementary Files.

### Supplemental Information
Supplemental information for this article can be found online at http://dx.doi.org/10.7717/peerj-cs.2096#supplemental-information.

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
