# Peer review of "A new integrated rough multi-criteria decision-making model for enterprise resource planning software selection"

_PeerJ Computer Science, doi:10.7717/peerj-cs.2096_

## Round 0.1 · original submission · Major Revisions

Based on the reviewers' comments and suggestions, this paper needs major revisions.

Reviewer 1 ·

Basic reporting

Well

Experimental design

Well

Validity of the findings

Well

Additional comments

Well

Reviewer 2 ·

Basic reporting

First, I must observe that this paper represents a revised version of the previously submitted paper which seems to have been rejected. I was not included in reviewing this paper in the initial version, so I can assess the new changed paper (with a rebuttal letter). I can confirm that the model is valid and well described, that revisions are clear and that the authors tried to adopt each proper comment provided by reviewers.

My opinion is that the study can contribute to the field of MCDM and the possibility of making proper decisions when selecting ERP software in different fields. The authors have developed a new extension of the WISP method for ranking alternatives. They have provided enough reasons, comparisons, and descriptions for their decision to present such a model. The paper can be considered for publication in PeerJ Computer Science because the paper falls within the scope of both aspects: field of application and proposed methods.

Experimental design

My comments are as follows:

1. Please explain why you have used a scale of 1-9 in this method.
2. You should discuss the results of BWM, which criterion is most important, and why?

Validity of the findings

My comments are as follows:

1. Please add references for the used methods in comparative analysis: (Rough MARCOS, Rough TOPSIS, Rough ARAS, Rough MOORA, Rough COPRAS, Rough SAW and Rough CODAS).

Additional comments

My comments are as follows:
1. Remove one of two almost the same keywords: ERP selection or ERP.
2. Full definitions of BWM and WISP are not provided in the abstract. Please define each abbreviation at the first time of application.
3. The paper contributions should be moved from the literature review to the introduction. It is usually to present contributions and novelties in the introduction rather than the literature review section.
4. Tables 8 and 9 need new titles. The results and the ranking aren't enough. Should be for example "The results of the BWM-WISP model" or something similar.

Reviewer 3 ·

Basic reporting

This article studies a new integrated multi-criteria decision-making model for ERP software selection based on a novel rough WISP and BWM methods. This paper uses rough BWM to determine criteria weights while newly developed rough WISP is used to rank ERP alternatives. It is interesting. I suggest major revision. Some comments are provided as follows.
The English writing should be polished with help of professionals since there are some typos and grammatical errors.

Experimental design

This paper uses rough BWM which is an extension of the classical BWM. But there is no any review on extensions of BWM in section 2. The authors are encouraged to conduct a more extensive review of relevant literature to enhance the current state of their work. It is worth noting that some noteworthy research articles in the fields of extension of BWM and MCDM have not been taken into account in the present form, such as Expert Systems with Applications, 236 (2024) 121213; Information Sciences, 666 (2024) 120404. It is necessary to make an overall review of literature to grasp the status of research.

Validity of the findings

The motivations should be elaborated. Why does this paper use rough BWM? What is the advantage of rough BWM?

Section 3.2 proposes the Rough WISP method. However, it just lists the steps. The reasonability and rationality are unknown. Please address them clearly.

Please add some solid comparative analyses in section 4 to strengthen the convincingness and credibility of this paper.

Additional comments

In sum, I recommend major revision.

---

## Round 0.2 · accepted · Accept

Based on the reviewers' comments and suggestions, this paper can be accepted for publication.

Reviewer 2 ·

Basic reporting

The authors have revised the manuscript as per the suggestions given by the reviewers. So I recommend this manuscript for publication.

Experimental design

no comment

Validity of the findings

no comment

Additional comments

In my opinion, the contributions of the paper, the quality of the research, and the presentation (text) of the research unequivocally recommend the paper for publication. I congratulate the authors on their quality paper.
The authors have revised the manuscript as per the suggestions given by the reviewers. So I recommend this manuscript for publication.

Reviewer 3 ·

Basic reporting

The manuscript has been revised well according to my previous comments. It can be accepted now.

Experimental design

See above.

Validity of the findings

See above.

Additional comments

See above.